# Phytotoxic Metabolites Produced by Fungi Involved in Grapevine Trunk Diseases: Progress, Challenges, and Opportunities

**DOI:** 10.3390/plants11233382

**Published:** 2022-12-05

**Authors:** Pierluigi Reveglia, Regina Billones-Baaijens, Sandra Savocchia

**Affiliations:** 1Institute for Sustainable Agriculture, CSIC, 14080 Córdoba, Spain; 2Gulbali Institute, Charles Sturt University, Locked Bag 588, Wagga Wagga, NSW 2678, Australia; 3School of Agricultural, Environmental and Veterinary Sciences, Charles Sturt University, Locked Bag 588, Wagga Wagga, NSW 2678, Australia

**Keywords:** *Vitis vinifera*, GTDs, phytotoxins, plant-pathogen interactions, omics

## Abstract

Grapevine trunk diseases (GTDs), caused by fungal pathogens, are a serious threat to vineyards worldwide, causing significant yield and economic loss. To date, curative methods are not available for GTDs, and the relationship between the pathogen and symptom expression is poorly understood. Several plant pathologists, molecular biologists, and chemists have been investigating different aspects of the pathogenicity, biochemistry, and chemical ecology of the fungal species involved in GTDs. Many studies have been conducted to investigate virulence factors, including the chemical characterization of phytotoxic metabolites (PMs) that assist fungi in invading and colonizing crops such as grapevines. Moreover, multidisciplinary studies on their role in pathogenicity, symptom development, and plant-pathogen interactions have also been carried out. The aim of the present review is to provide an illustrative overview of the biological and chemical characterization of PMs produced by fungi involved in Eutypa dieback, Esca complex, and Botryosphaeria dieback. Moreover, multidisciplinary investigations on host-pathogen interactions, including those using cutting-edge Omics techniques, will also be reviewed and discussed. Finally, challenges and opportunities in the role of PMs for reliable field diagnosis and control of GTDs in vineyards will also be explored.

## 1. Introduction

Grapevines are one of the most economically important crops worldwide, with approximately 48% of the world’s grape production used for wine production [1]. Fungal diseases are limiting factors to the production of wine grapes, impacting the quality of wine. Grapevine trunk diseases (GTDs), caused by one or several fungal pathogens, cause a progressive decline in vines resulting in a loss in productivity and eventual death of the vines [2]. Internal and external GTDs symptoms sometimes take several years to appear after infection; thus, they are considered slow-progression diseases [3]. However, over the past few decades, the frequency of symptoms reported due to GTDs has increased considerably [2,4]. This may be in part due to better identification methods being developed for the diseases and the causal pathogens. The relationship of GTDs with biotic and abiotic stresses, the expression of symptoms, and the lack of effective management strategies requires further investigation [2,4,5,6]. Current strategies for the management of GTDs rely mainly on remedial surgery and pruning wound treatments. Remedial surgery or trunk renewal involves the removal of the infected wood and an additional 10–20 cm of non-symptomatic wood from infected vines and water shoots to replace the missing vine or cordon [7,8]. Fungicide treatments applied to pruning wounds can also prevent infection by GTDs [9,10,11]. However, only a limited number of fungicides are commercially available and registered to manage GTDs due to some of the effective fungicides, such as sodium arsenite and benzimidazoles, being removed from the market due to human and environmental safety concerns [2]. Biological controls such as *Trichoderma*-based products [12,13,14,15] for managing GTDs have increased in popularity, particularly for organic and biodynamic vineyards where synthetic chemicals are not an option.

Up to 133 fungal species belonging to 34 genera have been associated with GTDs worldwide, although their role in the disease remains unknown for some [4]. The management of GTDs represents a major challenge for viticulturists, nurseries, technicians, and plant pathologists, mainly because of their complexity compared with other grapevine diseases. There are no cultivated or wild grapevine taxa that are known to be resistant to GTDs [5]. The varying symptoms caused by GTDs increase the complexity of accurately identifying them in vineyards, and in many cases, the disease is recognized only once it has been established in the grapevine. There may be many organisms interacting, and the infection by one virulent pathogen could lead to a weakened vine that is more vulnerable to infection by another [16]. These issues add further complexity in linking the symptoms with the causative pathogens. The main GTDs that threaten vineyards worldwide are Esca complex, Eutypa dieback (ED), and Botryosphaeria dieback (BD).

Secondary metabolites (SMs) produced by microorganisms are defined as not indispensable compounds for their normal development but may provide a targeted advantage in specific conditions and habitats. Fungal SMs may play a role in virulence and plant-pathogen interactions and may be linked to symptom development in infected plants [17]. They can be divided into four main chemical classes: polyketides, terpenoids, shikimic acid-derived compounds, and non-ribosomal peptides [18]. Some SMs can be classified as fungal phytotoxic metabolites (PMs), and they are usually divided into host-selective toxins (HSTs) and non-host-selective (NHSTs) toxins [18]. Information on their exact mode of action is limited, and understanding their role in virulence remains an important challenge for chemists, molecular biologists, and plant pathologists [19,20,21,22,23,24]. 

The PMs produced by pathogenic fungi of crops and forest trees vary in activity and may display anticancer, anti-inflammatory, antioxidant, anti-fungal, and phytotoxic effects when assayed under different conditions and on different hosts [25,26,27]. Thus, many investigations have been carried out on their isolation and chemical and biological characterization during the past decades. These metabolites may also be used in agriculture as biological control agents for weeds or specific plant diseases [17,26,27].

The enzymes and gene clusters involved in the biosynthesis of SMs in fungi have been studied intensively [28,29,30]. Studying the pathogenicity genes in an ecological context can provide important clues to their functions in the life cycle of the pathogens and their role during the infection process [31,32]. Comparative genomics and transcriptomics are important tools that could assist in the identification of the complete gene set for the target pathway of SMs. Several SM gene cluster prediction software has also been developed, which can aid this process considerably. Genomics and transcriptomic tools can be used to study the expression of SM gene clusters at various stages of infection [33,34]. Comparative genomics can also be utilized for identifying the biosynthetic gene clusters of a target group of SMs bearing structural similarities [24,35]. Despite the importance of GTDs, few molecular studies have been conducted to improve understanding of the biology and ecology of the fungal species involved in these diseases.

This review provides an illustrative overview of the biological and chemical characterization of PMs produced by fungi involved in ED, Esca complex, and BD. Moreover, multidisciplinary investigations on host-pathogen interaction, including those using cutting-edge Omics techniques, will be discussed. Finally, challenges and opportunities on how PMs may result in reliable methods to better assist in field diagnosis and control of GTDs in vineyards will also be explored. 

## 2. Phytotoxic Metabolites Produced by Fungi Involved in GTDs and Their Role in Grapevine Tissues

Many studies have been carried out to investigate the virulence factors that assist fungi in invading and colonizing important economic crops such as grapevines. Carbohydrate-active enzymes (CAZymes), peroxidases, effector proteins, cytochrome P450s, cellular transporters, and secondary metabolism, including toxin production, are only some of the studied virulence factors [18,36,37]. Several studies have been conducted on the isolation and characterization of PMs produced in vitro by pathogens involved in GTDs in the last decades, and some of these compounds are typical for a specific pathogen [27]. Although, only a few have tried to investigate their role in pathogenicity and symptom expression. The lack of studies may be due to the complexity of GTDs, and the pathogens involved [5,38]. In vitro experiments or experiments based on artificially inoculated vines in the glasshouse are often useful, although these can only provide a partial view of the role of PMs as virulence factors. The production of PMs may be related to the expression of foliar symptoms. The general hypothesis is that they are produced by the pathogen in the trunk. They can migrate without being catabolized or entirely detoxified in the leaves where they cause the common observed symptoms (Figure 1). This hypothesis was proposed for the first time by Tey-Rulh et al. [39] to explain the foliar symptoms produced in grapevines by *Eutypa lata.*

However, the foliar symptoms appear between three and eight years post-infection and have also been observed to vary from year to year, and PMs have never been detected in grapevine leaves [40,41]. Consequently, the role of PMs in the pathogenicity and virulence of fungi involved in Esca complex, ED, and BD remains poorly understood. The studies, both in vitro and in planta, investigating the role of PMs will be described in the following sections.

### 2.1. Phytotoxic Metabolites Produced by Fungi Involved in Esca Complex

Fungi involved in Esca complex disease include *Phaeomoniella chlamydospora*, 27 species of *Phaeoacremonium*, *Pleurostoma richardsiae*, and ten species of *Cadophora* (Table 1) [4,38,42,43,44].

Among the different *Phaeoacremonium* and *Cadophora* spp. occurring in Esca, *Phaeoacremonium minimum* and *Cadophora luteo-olivacea* are the most prevalent [2,4]. 

The above-mentioned fungi belong to the *Ascomycota phylum* and have been isolated from grapevine necrotic wood and identified by molecular techniques using actin and β-tubulin gene regions [42]. Finally, numerous basidiomycetous species have been isolated from grapevine wood showing Esca symptoms. These fungi belong to the genera *Fomitiporia, Inocutis*, *Inonotus*, *Fomitiporella*, *Phellinus*, and *Stereum*. However, their role in the symptomatology of the disease requires further study [5,38].

The draft genome sequence of both *P. minimum* and *Ph. chlamydospora* are available [52,53]. In both fungi, the genome was enriched in clusters associated with secondary metabolism, including those encoding polyketide synthetases (PKS), non-ribosomal peptide synthetases (NRPS), and also secreted proteins predicted to be putative plant CAZymes [54]. More recently, comparative genome analysis and high throughput transcriptome studies have been conducted on *Ph. chlamydospora* [55].

The first reported PMs (Figure 2) from *P. minimum* were the naphthalenone pentaketides derivative named scytalone (**1**) and isosclerone (**2**) together with 4-hydroxybenzaldehyde (**3**) identified by spectroscopic method, essentially NMR [56,57]. The metabolites were assayed on detached grapevine leaves, compound 1 caused chlorosis, rounded to irregular, interveinal, or marginal spots, while **2** caused large, coalescent chlorotic and necrotic spots followed by withering and distortion of the lamina. The most recent report of PMs isolated from the culture filtrate of *P. minimum* dates back to 2004 when Mansour et al. [58] investigated the neutral organic extract and the acid organic extract. The neutral organic extract yielded scytalone (**1**), isosclerone (**2**), 4-hydroscytalone (**4**), 2,4,8-trihydroxytetralone (**5**), 3,4,8-trihydroxytetralone (**6**), 1,3,8-trihydroxynaphthalene (**7**). The main compounds in the acidic extract were flaviolin (**8**), isolated with traces of 2-hydroxyjuglone (**9**). These compounds were assayed on grapevine callus and *Arabidopsis thaliana*. The isolated PMs were divided into tetralones, such as scytalone and isosclerone, which promote callus growth, and naphthoquinones, like 2-hydroxyjuglone and flaviolin, which inhibit growth [58]. Little information is known about the function of the naphthalenone pentaketides **1**–**9** in vine cells or tissues [57,59]. Their activity might be related to their oxidant property, especially their interaction with reactive oxygen species (ROS) produced by the plant during the defense response [60,61]. Thus, the presence of the typical tiger-stripe symptoms on leaves from grapevines infected with Esca may be related to physiological changes caused by toxic metabolites produced by the causal pathogens in the trunk. In this case, these changes would be a response of the vine to the disease [59]. When the PMs were isolated and identified from the pathogens involved in Esca and BD, this hypothesis was also extended to these two GTDs [59]. However, further investigations are needed to support this hypothesis and to correlate the production of PMs with foliar symptoms.

*Phaeomoniella chlamydospora* was also investigated and produced the PMs tyrosol (**10**), 1-O-methylemodine (**11**), 3-hydroxy-5decanolide (**12**), (S)-4-hydroxyphenyllactic acid (**13**), 3-(3-methyl-2-butenyloxy)-4-hydroxybenzoic acid (**14**) (Figure 2). They were isolated together with the already reported scytalone (**1**) and isosclerone (**2**), and p-hydroxybenzaldehyde (**3**). Compound **3** showed the highest activity, inhibiting the protoplast growth of 20% at 100 mM and 100% at 1 mM. The two organic acids (S)-4-hydroxyphenyllactic acid (**13**), 3-(3-methyl-2-butenyloxy)-4-hydroxybenzoic acid (**14**) were also active on protoplasts at 1 mM concentration. Moreover, the authors point out that the aromatic aldehyde function, in the ortho or para position, is always present in the structures of the active metabolites and could be related to their activity [57].

### 2.2. Phytotoxic Metabolites Produced by Fungi Involved in Eutypa Dieback

The fungus *Eutypa lata* is a diatrypaceous fungus that is considered the main causal agent of Eutypa dieback. Although 22 species of *diatrypaceae* have been isolated from a grapevine showing typical ED symptoms (Table 2) [4,62,63,64], only *E. lata* is known to cause the typical foliar symptoms associated with this GTD [2,40,64].

These fungi have been isolated from asymptomatic tissues several centimeters ahead of lesion margins, indicating pathogen latency. *Eutypa lata,* among other GTD pathogens, was also found in asymptomatic grapevines aged 40 years or older, highlighting that there could be a balance between the plant microbiome and pathogenic fungi, assisting in preventing the development of the disease [70]. 

The first draft genome sequence of *E. lata* (UCR-EL1) was published in 2013 [71]. The study provided a preliminary inventory of the potential virulence factors and an abundant repertoire of cell wall-degrading enzymes, and a high number of putative cytochrome P450 monooxygenases implicated in lignin oxidation [71]. In 2015, Morales-Cruz and co-authors highlighted a great expansion of families of genes involved in the biosynthesis of toxins, including polyketide synthesis (t1PKS) [72]. More recently, the first whole-genome sequencing and comparative genomics study on 40 *E. lata* isolates from Australia was reported [73]. 

The first investigation on PMs produced by *E. lata* was carried out in 1989 [74], where the screening of the bioactive organic fractions led to the isolation of eight new aromatic compound metabolites characterized by a 3-methylbut-3-en-1-ynyl functional group: eutypine (**15**), eutypinol (**16**), O-methyleutypine (**17**), O-methyleutypinol (**18**), eutypune carboxylic acid (**19**), 4-ydroxy-3,4-dihydroxy-3-methylbut-l-ynyl)benzyl alcohol (**20**), 4-hydroxy-3-(3,4-dihydroxy-3-methylbut-l-ynyl)benzaldehyde (**21**), 3-hydroxy-3,4-diliydroxy-3-methylbut-I-ynyl)benzoic acid (**22**) (Figure 3). Moreover, the authors detected 5-formyl-2-(methylvinyl)[1]benzofuran (**23**) in the culture medium, which is also obtained from compound 1 under middle acid condition (Figure 3). The compounds were tested on tomato seedlings and grapevine leaves. Eutypine (**15**) showed the highest phytotoxic activity. Moreover, the authors suggested that the aldehyde group and a free OH in para-position could be necessary for the activity [75].

In the same year, the same authors isolated novel allenic epoxycyclohexanes 5-(3-methylbuta-l.3-dienylidene) -2,3-epoxycyclohexane-l,4-diol (**24**), 6-hydroxy-2,2-dimethyl-5,6.7,8-tetrahydro-7,8-epoxychroman (**25**) and 8-hydroxy-2,2-dimethyl-5.6,7,8-tetrahydro-6.7-epoxychroman (**26**) (Figure 2), together with the already reported Eutypine (**15**). Their structures were established by a combination of spectroscopic, X-ray, and chemical modifications. Again, the most active compound was Eutypine (**15**). Finally, the authors suggested that compound **24** could be the key intermediate in the transformation of acetylenic compounds into tetrahydrochromanone derivatives [74].

Eutypoxide B (**27**), biogenetically related to 5-(3-methylbuta-l.3-dienylidene)-2,3-epoxycyclohexane-l,4-diol (**24**), was isolated from the culture filtrate of *E. lata*. The structure was confirmed by NMR and X-ray. The authors also conducted a total synthesis of **27,** revealing that the key intermediate for the synthesis is a cyclohexanecarbaldehyde compound [76].

Molyneux and co-authors studied three strains of *E. lata* and their metabolites produced in artificial media by HPLC and GC MS [77]. The novel metabolites eulatinol (**28**), which is structurally related to the already known siccayne (**29**), and eulatachromene (**30**) (Figure 3), a novel chromene analog, were identified by spectroscopic methods as a methoxyquinol derivative. Only one strain produced eutypine (**15**) in a low amount, whereas the primary metabolite was the corresponding alcohol, eutypinol (**16**). In the bioassay on grapevine leaf discs, neither eutypinol (**15**) nor siccayne (**29**) showed phytotoxicity, whereas eulatinol (**28**) and eulatachromene (**30**) caused necrotic spots [77].

Several attempts have been made to identify eutypine from different tissues showing ED symptoms [41,78,79]. In vitro cultures of *E. lata* isolates produced various SMs, of which eutypine was the main metabolite. However, HPLC analysis of extracts from wood, shoots, and leaves exhibiting symptoms of dieback failed to show the presence of any metabolites [78]. Micropropagated grapevine plantlets treated with crude or purified culture filtrates from nine isolates of *E. lata* grown on malt yeast broth resulted in various SMs being identified. However, no single compound was consistently detected [79]. A derivative of eutypine, eutypinol, was detected in micropropagated grapevine plantlets inoculated with *E. lata* mycelium, but PMs were not detected in the sap of vines that had been artificially inoculated with the pathogen [79]. The reasons for the unsuccessful detection of eutypine remain unclear. However, it is possible that following their production; such metabolites are rapidly broken down into compounds that cannot be detected by HPLC [78]. Alternatively, the translocation of metabolites does not occur because the compounds are sufficiently reactive damaging plant tissues in the proximity of fungal infection, reacting in such a way that they are bound irreversibly. Finally, the authors of these studies suggested that as phenolic compounds, the metabolites would be susceptible to oxidative polymerization by plant phenol oxidases, possibly accounting for the dark, wedge-shaped areas typical of *E. lata* infection [41,78,79].

### 2.3. Phytotoxic Metabolites Produced by Fungi Involved in Botryosphaeria Dieback

Botryosphaeriaceae have a cosmopolitan distribution and occur on a wide range of annual and perennial hosts, including grapevines [80,81,82]. They have been described as endophytes or latent pathogens causing serious diseases [82,83]. However, the status of Botryosphaeriaceae species as endophytes in grapevines remains unclear [84]. Botryosphaeriaceae species occur in grape-growing regions of Africa, Asia, Australia, Central America, Europe, and South America [51,85,86,87]. Different species of Botryosphaeriaceae belonging to the genera *Botryosphaeria*, *Diplodia*, *Dothiorella*, *Lasiodiplodia*, *Neofusicoccum*, *Neoscytalidium*, *Phaeobotryosphaeria* and *Spencermartinsia* have been reported to be associated with BD of grapevines worldwide. The isolation of *S. westrale*, *S. plurivora*, *Do. neclivorem*, *Do. vineagemmae* and *Do. vidmadera* from perennial cankers has brought the total number of Botryosphaeriaceous species isolated from grapevines worldwide to 40 (Table 3) [6,88,89,90,91]. The species infecting grapevines can be classified according to their virulence and can be divided into three different groups, including the highly virulent species, *Lasiodiplodia* spp. and *Neofusicoccum* spp., intermediately virulent *B. dothidea* and *Diplodia* spp., while *Dothiorella* spp. and *S. viticola* are weakly virulent [84].

Low genetic variability is reported from many geographical locations for *Botryosphaeria*, *Diplodia*, and *Lasiodiplodia* species [102]. The first draft of the *N. parvum* genome was published in 2013, while the first draft genome sequence of *D. seriata* isolate F98.1 was obtained by Siegwald and co-authors [103]. Studies focused on genetic diversity, the evolution of possible virulence factors, and the pathogenicity of Botryosphaeriaceae species have demonstrated variable levels of virulence between species and isolates [92,93,100,101,104]. All these pathogens have gene clusters involved in the production of carbohydrate-active enzymes (CAZymes), peroxidases, cytochrome P450s, cellular transporters, and secondary metabolism, with a particular focus on toxin production [54,72]. 

Studies on the phytotoxicity of PMs produced by Botryosphaeriaceae spp. involved in GTDs were first reported by Martos, Andolfi et al. [105]. Species isolated from declining grapevines in Spain (*B. dothidea, D. seriata, Do. viticola, N. luteum and N. parvum*) produced hydrophilic high-molecular-weight compounds, exopolysaccharides (EPSs) with phytotoxic properties in liquid culture. In addition, lipophilic low molecular weight phytotoxins were isolated from the organic extracts of the culture filtrates of *N. luteum*, and *N. parvum* [105].

Among the Botryosphaeriaceae spp. infecting grapevines worldwide, *N. parvum* and *D. seriata* are, respectively, the most aggressive and the most widespread worldwide [84], and therefore they are the most studied species producing PMs. Several phenolic dihdroisocoumarins, such as (*R*)-mellein (**31**), (3*R*,4*R*)-and (3*R*,4*S*)-4-hydroxy melleins (**32**,**33**) and 5-hydroxymellein (**34**) (Figure 4) were identified from the organic extract of the culture filtrate of *D. seriata* [106]. Furthermore, an unknown mellein characterized by NMR was identified as (3*R*,4*R*)-4,7- dihydro9mellein (**35**, Figure 4).

For *N. parvum*, four phytotoxic metabolites were isolated from organic extracts and identified by spectroscopic and physical examination as isosclerone (**2**), tyrosol (**10**, Figure 2), and the previously reported 4-hydroxy-mellein cis and trans mellein (**32**, **33** Figure 4) [107]. Liquid chromatography-diode array screening of the organic extract of the cultures of 13 isolates of *N. parvum* resulted in 13 compounds belonging to four different chemical families being identified through spectroscopic analyses and by comparison to previously published literature as detailed in Figure 3: (*R*)-mellein (**31**), (3*R*,4*R*)-and (3*R*,4*S*)-4-hydroxy melleins (**32**, **33**), (+)-terremutin (**36**), (+)-terremutin hydrate (**37**) (+)-epi-sphaeropsidone (**38**), (-)-4-chloro-terremutin hydrate (**39**), (+)-4—hydroxysuccinate-terremutin hydrate (**40**), (6*R*,7*R*)-asperlin (**41**), (6*R*,7*S*)-dia-asperlin (**42**), (*R*)-3-hydroxymellein (**43**), 6-methyl-salicylic acid (**44**), 2-hydroxypropyl salicylic acid (**45**) [108].

(*R*)-mellein and its derivatives belong to the class of isocoumarines, a class of natural compounds well known to have a wide range of biological activity [109,110]. Moreover, (*R*)-mellein is a typical PM produced by Botryosphaeriaceae spp. involved in GTD and is phytotoxic to grapevine leaves at different concentrations [106,107].

Ramirez-Suero et al. showed that (*R*)-mellein could not explain the toxicity of the extracellular organic extract of *D. seriata* and *N. parvum* [111]. Purified (*R*)-mellein was added to the culture medium of calli, but only delayed necrosis and a lower-level expression of defense genes was observed. In addition, the extracellular compounds from *N. parvum* appeared to be more toxic than those produced by *D. seriata*. Finally, the authors suggested that it is possible that the pathogenicity of these two fungi depends on synergistic action between the secretion of other types of PMs, such as derivatives of mellein or high molecular weight phytotoxins such as polypeptides or EPSs [111].

The extracellular EPSs produced by an isolate of *N. parvum* isolated from infected grapevine wood in a vineyard in Spain were biologically and chemically characterized by Cimmino et al. [112]. The EPS was characterized as a mannan having a backbone consisting of (1→6)-linked mannopyranose units, almost all branched at the 2nd position, whereby the arms were composed of 2- and/or 3- linked units. The phytotoxic activity was observed when assayed on grapevine leaves. However, the three replicates of each tested concentration developed symptoms at different times, and differences in the type of symptoms induced were observed; therefore, a conclusion could not be drawn [112].

More recently, purified secreted proteins by *N. parvum* and *D. seriata* were assayed on suspension cells of two different *Vitis* genotypes (*V. rupestris* and *V. vinifera* cv. Gewurztraminer) with putative varying susceptibility to BD [113]. The Vitis cells were able to detect secreted proteins produced by Botryosphaeriaceae and respond by producing ROS and by the production of reactive oxygen species and prompt alkalinization of the extracellular medium. *Vitis rupestris* is characterized by higher medium alkalinization, cell death, and more intense induction of pathogenesis-related genes, whereas V. vinifera cv. Gewurztraminer produced a higher amount of antifungal compound δ-viniferin. The results further suggested that even if the grapevine can react rapidly to BD pathogens, the defense responses are most likely not strong enough to restrict the growth of the pathogen. However, further studies are required to determine the sequences of the secreted proteins and their mode of action [113].

*Neofusicoccum australe* involved in grapevine decline in Sardinia produced a new cyclohexenone oxide, namely, cyclobotryoxide (**46**, Figure 5), that was isolated together with 3-methylcatechol (**47**, Figure 4) and tyrosol. Cyclobotryoxide was the most active metabolite in the different bioassays performed [114].

Several endophytic and pathogenic fungi can produce PMs that are also biosynthesized by their host plants [19]. The GTD pathogens, *Lasiodiplodia* spp., are capable of producing jasmonic acid (**48**, Figure 5), a known plant hormone, and some of its derivatives when grown in in vitro conditions. Jasmonic acid, its methyl ester (**49**, Figure 5), and Lasiojasmonate A-C (**50**-**52**, Figure 5) were isolated from the grapevine pathogen *L. mediterannea* [115]. The mode of action of jasmonic acid and lasiojasmonate A as fungal phytotoxins were investigated, and the results suggested that the production of a jasmonic acid derivative such as lasiojasmonate A occurs during the late stages of infection to induce plant jasmonic acid responses such as cell death and to facilitate fungal infection [20,116]. 

More recently, two novel compounds identified as lasiolactol A and B were isolated and characterised (**53**, **54**
Figure 5) from another strain of *L. mediterranea* isolated from grapevines in Sicily. These two novel molecules were isolated together with botryosphaeriodiplodin (**55**, Figure 4), (5R)-5-hydroxylasiodiplodin (**56**, Figure 5), and (3S,4R,5R)-4-hydroxymethyl-3,5-dimethyldihydro-2-furanone (**57**, Figure 5), all previously characterized secondary metabolites [117]. 

LC-MS analysis of *L. brasiliense*, *L. crassispora*, *L. jatrophicola*, *L. pseudotheobromae* isolated from grapevines with BD in Brazil produced jasmonic acid, and *L. brasiliense* also produced (3*R*,4*S*)-4-hydroxymellein (**33**) [118]. *Lasiodiplodia euphorbicola* produced (*R*)-mellein (**31**), (3*R*,4*R*)- and (3*R*,4*S*)-4-hydroxymellein (**32**, **33**) and tyrosol (**10**), while *L. hormozganensis* synthesized tyrosol (**10**) and p-hydroxybenzoic acid (**58**, Figure 5). All these compounds were phytotoxic when assayed on grapevine leaves [118]. More recently, (*R*)-mellein (**31**) and tyrosol (**10**) were also isolated from *L. laeliocattleyae* (syn. *egyptiacae*) from Brazil, further confirming that compound **31** is typically produced by Botryosphaeriaceae species [119].

In the same year, 24 fungal isolates representing the eight most widespread and virulent Botryosphaeriaceae involved in BD in Australia were investigated for the first time for their ability to produce phytotoxic metabolites [120]. The chromatographic profiles of the organic extracts of isolates of *D. seriata*, *D. mutila*, *N. parvum*, *N. australe*, and for the first time, those of *N. luteum*, *Do. vidmadera* and *S. viticola* showed that these fungi were able to produce several and different metabolites. 

From the organic extract of *S. viticola* DAR78870, two novel compounds named spencertoxin (**59**, Figure 6) and spencer acid (**60**, Figure 6) were isolated. Both Compound **59,** which was spectroscopically characterized as adipyridine-butane-1,4-diol, and compound **60,** a new diacrylic acid derivative, were isolated. Moreover, the already known 4-hydroxybenzaldehyde (**3**), and 2-(4-hydroxyphenyl) acetic acid (**61**, Figure 6) were also isolated and spectroscopically dereplicated. Spencertoxin (**59**), p-hydroxybenzaldehyde (**3**) and 2-(4-hydroxyphenyl) acetic acid (**61**) showed differences in phytotoxicity when tested on grapevine leaves of *V. lambrusca* and *V. vinifera* cv. Shiraz [121].

From *Do. vidmadera* DAR78993, six polyphenols were isolated. One was a novel polyphenol characterized by 1D and 2D 1H, and 13C NMR spectroscopy, as 5-hydroxymethyl-2-isopropoxyphenol (**62**, Figure 6). The other polyphenols isolated and dereplicated were: tyrosol (**10**), benzene-1,2,4-triol (**63**, Figure 6), resorcinol (**64**, Figure 5), 3-(hydroxymethyl) phenol (**65**, Figure 6), and protocatechuic alcohol (**66**, Figure 6). Compound 66 was the main metabolite. The compounds showed similar phytotoxic effects to one another when assayed on tomato seedlings. However, when assayed on grapevine leaves (*V. vinifera* cv Shiraz), resorcinol (**64**) was the most toxic compound. This was the first time these polyphenols, naturally occurring in other microorganisms and plants, have been isolated from fungal organisms involved in GTDs [122].

Diploquinones A and B (**67**, **68**, Figure 6) were isolated together with vanillic acid (**69**, Figure 6) from the organic extract of *D. mutila* DAR78993. Compounds **67** and **68** were characterized by 1D and 2D 1H and 13C NMR as 6,7-dihydroxy-2-methoxy-5-methylnaphthalene-1,4-dione and 3,5,7trihydroxy-2-methoxynaphthalene-1,4-dione, respectively. Moreover, vanillic acid (**69**) was isolated for the first time as a fungal phytotoxin and as a metabolite from *D. mutila*. A bioassay was conducted at different concentrations on detached grapevine leaves, with vanillic acid (**69**) resulting in the highest phytotoxic effect. Compounds **67** and **68** showed varying degrees of toxicity depending on the tested concentration [123].

Several phytotoxic metabolites were isolated from the organic extract of *N. australe* DAR79506 and *N. parvum* DAR80004 and, for the first time, from *N. luteum* DAR81016. A novel disubstituted furo-α-pyrone, luteopyroxin (**70**, Figure 6), a hexasubstituted anthraquinone, neoanthraquinone (**71**, Figure 6), and a trisubstituted oxepi-2(7H)-one, luteoxepinone (**72**, Figure 6), respectively, together with the known (±)-nigrosporione (**73**, Figure 6), tyrosol (**10**), (*R*)-mellein (**31**), and (3*R*,4*S*)- and (3*R*,4*R*)-4-hydroxymellein (**32**, **33**) was characterized from the culture filtrate of N. luteum. The three melleins and tyrosol were also produced by *N. parvum*, while *N. australe* produced (*R*)-mellein (**31**), tyrosol (**10**), neoanthraquinone (**71**), and 4-cresol (**74**). All the compounds were identified and characterized by 1D and 2D 1H and 13C NMR and high-resolution electrospray ionization mass spectrometry. Moreover, the relative and absolute configurations of compound **69** were determined by nuclear Overhauser effect spectroscopy and experimental and calculated electronic circular dichroism data. Neoanthraquinone (**71**) showed the highest toxic effect, causing severe wilting when assayed on grapevine leaves. Compounds **70**, **72,** and **73** showed different degrees of toxicity depending on the concentration assayed [124].

The relationship between the Botryosphaeriaceae fungi found in diseased wood, and the expression of foliar symptoms observed in some cultivars in the northern hemisphere, remains unclear. A recent investigation by Reis et al. [125] resulted in the development of a simple model system to reproduce the foliar symptoms caused by *D. seriata* and N*. parvum* to better characterize fungal pathogenicity and to determine the mechanisms involved in symptom development. The experiment was repeated four times from 2011 to 2014, and eight months after inoculation, the percentage of plants showing foliar symptoms was greater for vines inoculated with *N. parvum* than those inoculated with *D. seriata*. This study is the first to report the reproduction of foliar symptoms in inoculated vines with a frequency close to those observed in the vineyard. Furthermore, the authors monitored the expression of plant stress-related response genes and observed an upregulation of PR6 (PR6, chitinase, and β-1,3-glucanase) in the leaves. However, these data, together with other investigations [111,126,127], suggest that the natural defense responses produced by the plant are inefficient against these pathogens. Understanding the mechanisms involved in the infection of grapevine by Botryosphaeriaceae is important in developing strategies that limit the spread of the disease.

The infected trunk wood and green shoots from four grapevines (cv. Chardonnay; Avize, Epernay, Champagne-Ardenne region, France and Mourvèdre; Rodilhan, Nîmes, Languedoc Roussillon region, France) showing Esca and/or BD foliar symptoms were analyzed to determine whether the common PMs identified in in vitro cultures of *N. parvum* were also produced in planta [108]. The analysis was performed for (*R*)-mellein and (+)-terremutin, which were considered representatives of two of the isolated chemical families, through HPLC-DAD-MSn using organic extracts from powdered wood. Both PMs were detected in the extracts of brown-striped and black-streaked wood samples. This is the only report of PMs produced by N. parvum in grapevines showing Esca or BD in vineyards [108]. However, as also described by the authors, the small number of samples analyzed and the lack of information on which pathogens were present in the infected analyzed wood does not allow a definitive conclusion to be drawn on the role of PMs in the disease.

More recently, wood samples showing BD symptoms and one-year-old vines inoculated with *D. seriata*, *S. viticola*, and *Do. vidmadera* were analyzed by fungal isolations, quantitative PCR (qPCR), and targeted LC-MS/MS to detect three PMs: (*R*)-mellein, protocatechuic acid, and spencertoxin [128]. The key to the study was combining conventional plant pathology, molecular biology, and analytical chemistry to gain insight into the development of foliar symptoms. Only (*R*)-mellein was detected in symptomatic naturally-infected wood and vines artificially-inoculated with *D. seriata*; however, it was absent in all non-symptomatic wood. Under experimental conditions, (*R*)-mellein was not translocated into other parts of the vine and may be produced by the pathogen during infection to break down the wood. This study highlighted that the simple presence of PMs in the wood might not be enough to induce foliar symptoms in grapevines with BD. These symptoms may arise from interactions between biotic and abiotic stresses, which require more in-depth studies. Nevertheless, the amount of (*R*)-mellein detected was correlated with the amount of pathogen DNA detected by qPCR. Moreover, a comparison of the amount of (*R*)-mellein detected in other wood samples artificially inoculated with *N. parvum* showed a higher amount six months post-inoculation than in vines infected with *D. seriata* at 12 months post-inoculation. These findings further support those previously reported that the amount of (*R*)-mellein produced by *N. parvum* and *D. seriata* under in vitro conditions is proportional to the aggressiveness of the pathogens [111].

Two *N. parvum* isolates and a UV mutant were studied for their phytotoxin production in vitro, their pathogenicity on the grapevine, and their genome sequence [129]. Essentially (*R*)-mellein and (-)-terremutin were the main phytotoxins produced by the selected isolates. However, the variation in the amounts produced did not affect the pathogenicity of the isolates. High polymorphisms in genes involved in (-)-terremutin and (*R*)-mellein biosynthesis were identified, which could be a result of the evolution and adaptation of the pathogen to the environment. The study on grapevine immunity indicated that neither (-)-terremutin nor (*R*)-mellein alone is essential for the pathogenicity of *N. parvum* as the isolate/mutant not producing PMs in vitro was also pathogenic. Alternatively, (*R*)-mellein and (-)-terremutin could act as a virulence factor (i.e., quantitatively increasing the disease) than as a pathogenicity factor (i.e., required to trigger the disease). To confirm this hypothesis for (*R*)-mellein, mutants unable to produce PMs should be created, and their pathogenicity should be compared with their wild-type parent [129].

## 3. Metabolomics Studies 

Authors Metabolomics is the latest of the ‘omics sciences,’ and its relevance in plant science has increased in the last decades [130,131,132,133]. This ‘omics science’ has been applied to gain biological information on the host-pathogen interaction, on the process of abiotic stress, or it has been applied to identify potential biomarkers for a specific stress condition [134,135]. Few metabolomics studies have been carried out on GTDs, with most studies focused on the plant defense response to the colonization of the host by the pathogen rather than the phytotoxic metabolites produced by the fungi during infection [136,137,138]. 

Co-cultivation of fungal species has emerged as a promising method to discover novel antimicrobial metabolites by activating biosynthetic pathways in the selected fungal species. An untargeted metabolomics approach was applied to the co-culture of *E. lata* and *Botryosphaeria obtusa* (syn. *D. seriata*) to shed light, at a molecular level, on microbial inter- and intra-species cross-talk [139]. *Eutypa lata* and *B. obtusa* were grown in vials, and volatile and non-volatile constituents were profiled over nine days by headspace solid-phase microextraction gas chromatography-mass spectrometry (HS-SPME-GC-MS) and by liquid chromatography high-resolution mass spectrometry (LC-HRMS), respectively. For a comprehensive evaluation of the complex LC-HRMS and GC-MS datasets, a chemometrics analysis was combined with an Analysis of Variance and Partial Least Squares approach, namely AMOPLS. 2-nonanone (**75**, Figure 7) was incrementally produced over time in the volatile fraction and was found to be co-expressed with O-methylmellein (**76**, Figure 7), detected in the organic extract. The induction dynamics of **75** and **76** were similar, and their amount increased over the time of the analysis. In addition, the antifungal activity of O-methylmellein (**76**) is already known, and the authors assayed 2-nonanone (**75**), demonstrating its antifungal activity against *B. obtusa* and *E. lata*. Unfortunately, it was not possible to determine if 2-nonanone was produced by a single fungal species or both because **75** was not detected in the single-culture experiments. In conclusion, these outcomes could assist in deciphering the interaction between fungal communities and the various events that can trigger a specific biochemical response [139].

More recently, the first study of both untargeted and targeted LC-MS/MS-based metabolomics to investigate differences in the metabolic composition of organic extracts of three *Phaeoacremonium* species involved in Esca was reported [140]. Several isolates of *P. italicum*, *P. alvesii*, and *P. rubrigenum* were grown in vitro, and the culture filtrates, and organic extracts, were assayed for phytotoxicity. The phytotoxicity varied within and among the culture filtrates, and a toxicity score was assigned. The culture filtrate was extracted at two different pHs, and UHPLC-Orbitrap was used for metabolic profiling. Following data pre-treatment and multivariate statistical analysis (PCA and PLS-DA), the dereplication of already known compounds was conducted by matching retention times and HRMS and MS/MS data with those in online databases. Scytalone (**1**) and isosclerone (**2**) were recognized in the regular pH organic extract. The identity of these PMs was further confirmed by comparison of their spectrometric data with those of pure standards, according to the recommendation of the Metabolomics Standard Initiative [141]. To quantify scytalone (**1**) and isosclerone (**2**), a sensitive multi-reaction monitoring (MRM) method was developed on a UPLC-QTrap. The absolute quantification revealed that the amount of (**1**) and (**2**) differed within and among fungal species. Finally, further analysis is needed to identify the unknown metabolites detected in the untargeted analysis [140].

## 4. Challenges and Perspective

The more information we know about the secondary metabolism of a pathogen, the more we can understand its life cycle. Differences in the phytotoxicity of the isolated compounds indicate that not all the SMs produced by a fungal pathogen play a role in the development of disease symptoms. However, they may still play a significant role in the plant-pathogen interaction. Future investigations on how these compounds interact with host cells will assist in understanding their mode of action. The assignment of the chemical structure to a fungal phytotoxin is important to understand its mode of action and function in plant-pathogen interactions. Currently, Computer-Assisted Structure Elucidation (CASE) programs can assist in assigning structures to complex natural products. The use of CASE has reduced the risk of the wrong assignment by generating structures, ranked by probability, that are consistent with the input data [142]. Moreover, advances in bioinformatics tools have accelerated the discovery of previously undescribed natural product molecules. MS/MS molecular networking has emerged as a promising approach to dereplicate complex natural product mixtures, accelerating the discovery of novel bioactive compounds [143]. Finally, the last decades have also seen very rapid advances for in situ chemical analysis. Many of those techniques rely on the power of mass spectrometry using various ionization sources, from DESI to LAESI [144]. This, in turn, could assist in developing more specific and effective control methods for plant diseases and in designing strategies to alter the balance of these interactions to decrease pathogenicity. 

Bioactive metabolites from fungi can show a wide range of biological activities [25,26,27]. For this reason, it may be valuable to assay other biological activities of the isolated compounds. For instance, diploquinone A and B and neoanthraquinone could be screened for different activities (antibiotic, antimicrobial, anticancer, etc.) because it is well-known that compounds with a quinone and anthraquinone backbone show a wide range of biological activities [60,145,146]. Finally, data from this research could be useful in developing a database of all compounds produced by the corresponding species. This database could assist future investigations for identifying potential biomarker(s) of BD. 

Studies on the role of PMs in the pathogenicity and virulence of GTDs pathogens, using molecular and analytical chemistry techniques, showed that the mere presence of PMs in the wood might not be sufficient to result in the development of GTDs symptoms. The development of symptoms could be more complex than previously thought and may be due to the interaction of biotic and abiotic stress, such as water stress, drought, and heat stress. There are limited studies investigating the response to combined stresses. Nevertheless, multiple factors can impact the response and recovery of the grapevine, such as the intensity, duration, and timing of the stresses [147]. Recently, Patanita et al. also showed the importance of the grapevine fungal community, and not only of GTDs-associated fungi, in the expression of GTDs symptoms [148]. Therefore, the grapevine response to a combination of abiotic and biotic stresses is complex and cannot be assumed from the response to every stress. Thus, it is very likely that a diseased and healthy grapevine will show differences in metabolic fingerprints. Investigating the production of PMs and their role in the progression of diseases in open field conditions requires careful experimental design. Primarily, the research should have a longitudinal design to cover seasonal changes. In addition, other key factors to consider are (i) the selection of variables (temperature, rainfall, soil composition, microbial community) to be measured during the experiment; (ii) the selection of vineyards located in different climatic zones; (iii) an appropriate number of biological replicates for any sampling period and location according to the selected statistical power.

Investigations using complex samples, such as wood extracts obtained under different stress conditions, may result in the identification of specific metabolic fingerprints that could be related to specific processes in plants, such as defense mechanisms or stress responses. Using this framework, multi-omics analysis, integrating multiple omics platforms, may provide a consistent holistic view deepening the biological, biochemical, and biophysical understanding of fungal pathogens [149]. For instance, genomics could be used for the identification of genes markers and/or genes involved in PMs biosynthesis, dual transcriptomics for studying virulence factors and host-pathogen interactions [150], proteomics to investigate the regulation of signal transduction pathways, and, lastly, metabolomics to research the temporal regulation and to characterize potential effector compounds [151]. Metabolomics and dual transcriptomics could be useful tools to study, at the same time, the grapevine response and the fungal colonization process under different abiotic stress conditions that are more aligned with field growing conditions [134,135]. Nevertheless, it is important to highlight that protocol standardizations and subsequent multivariate analysis, reliability, and reproducibility of extractions and other pre- and post-analytical steps, are still a bottleneck for OMICS techniques [141,152]. The definition of best practices and harmonization of all these steps are fundamental to improving reproducibility and avoiding misleading biological results.

The search for effective protectants that can provide long-term protection against GTDs remains a challenge to plant pathologists worldwide. Recent studies showed nano-particle-based products such as nanocarriers for fungicides and RNA-interference molecules offer a better option as these products are less toxic and have a longer shelf-life, and are therefore less harmful to the environment [153,154,155]. Multifaceted studies which include the role of secondary metabolites in pathogenesis, combined with nanotechnology, could be noteworthy for developing more sustainable and specific control methods as alternatives to chemical-based approaches [156,157,158].

## Figures and Tables

**Figure 1 plants-11-03382-f001:**
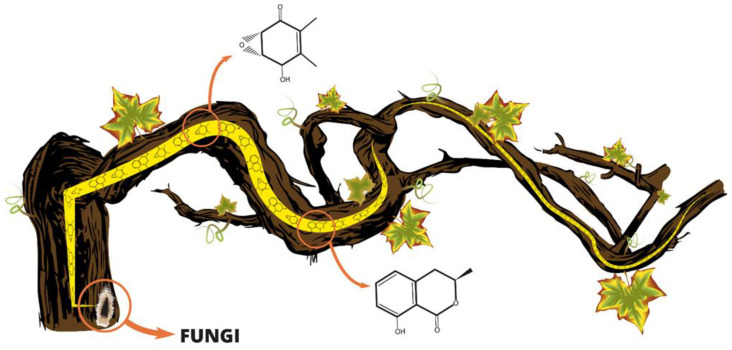
Schematic representation of the hypothesis that fungal phytotoxins are able to translocate within a vine resulting in foliar symptoms.

**Figure 2 plants-11-03382-f002:**
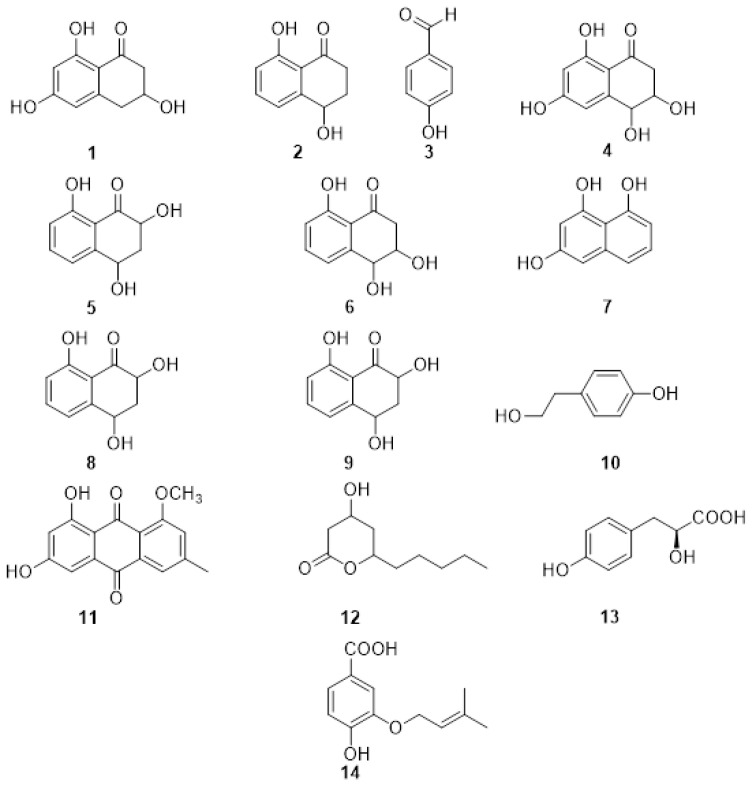
Structure of scytalone (**1**), isosclerone (**2**), 4-hydroxybenzaldehyde (**3**), 4-hydroscytalone (**4**), 2,4,8-trihydroxytetralone (**5**), 3,4,8-trihydroxytetralone (**6**), 1,3,8-trihydroxynaphthalene (**7**), flaviolin (**8**), 2-hydroxyjuglone (**9**), tyrosol (**10**), 1-O-methylemodine (**11**), 3-hydroxy-5decanolide (**12**), (S)-4-hydroxyphenyllactic acid (**13**), 3-(3-methyl-2-butenyloxy) -4-hydroxybenzoic acid (**14**).

**Figure 3 plants-11-03382-f003:**
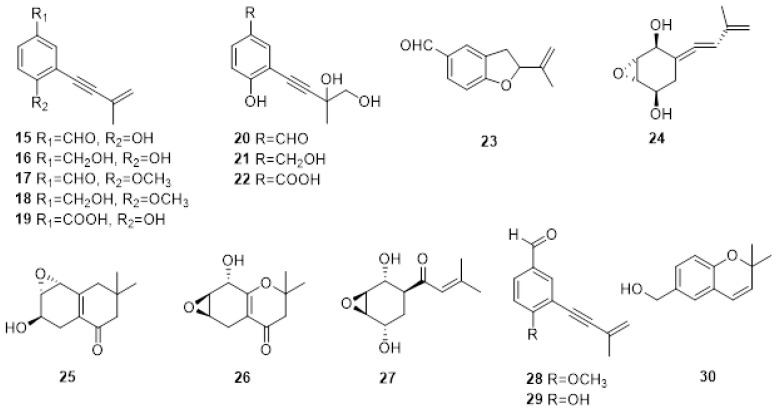
Structure of scytalone eutypine (**15**), eutypinol (**16**), O-methyleutypine (**17**), O-methyleutypinol (**18**), eutypune carboxylic acid (**19**), 4-ydroxy-3,4-dihydroxy-3-methylbut-l-ynyl)benzyl alcohol (**20**), 4-hydroxy-3-(3,4-dihydroxy-3-methylbut-l-ynyl)benzaldehyde (21), 3-hydroxy-3,4-diliydroxy-3-methylbut-I-ynyl)benzoic acid (**22**), 5-formyl-2-(methylvinyl)[1]benzofuran (**23**), 5-(3-methylbuta-l.3-dienylidene)-2,3-epoxycyclohexane-l,4-diol (**24**), 6-hydroxy-2,2-dimethyl-5,6.7,8-tetrahydro-7,8-epoxychroman (**25**) and 8-hydroxy-2,2-dimethyl-5.6,7,8-tetrahydro-6.7-epoxychroman (**26**), eutypoxide B (**27**), eulatinol (**28**), siccayne (**29**), eulatachromene (**30**).

**Figure 4 plants-11-03382-f004:**
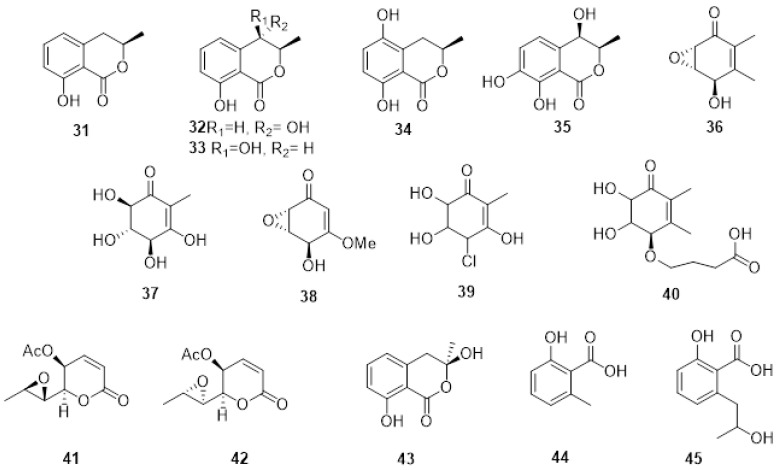
Structure of: (*R*)-mellein (**31**), (3*R*,4*R*)-and (3*R*,4*S*)-4-hydroxy melleins (**32**, **33**), 5-hydroxymellein (**34**), (3*R*,4*R*)-4,7- dihydroxy mellein (**35**), (+)-terremutin (**36**), (+)-terremutin hydrate (**37**), (+)-epi-sphaeropsidone (**38**), (-)-4-chloro-terremutin hydrate (**39**), (+)-4—hydroxysuccinate-terremutin hydrate (**40**), (6*R*,7*R*)-asperlin (**41**), (6R,7S)-dia-asperlin (**42**), (R)-3-hydroxymellein(**43**), 6-methyl-salicylic acid (**44**) and 2-hydroxypropyl salicylic acid (**45**).

**Figure 5 plants-11-03382-f005:**
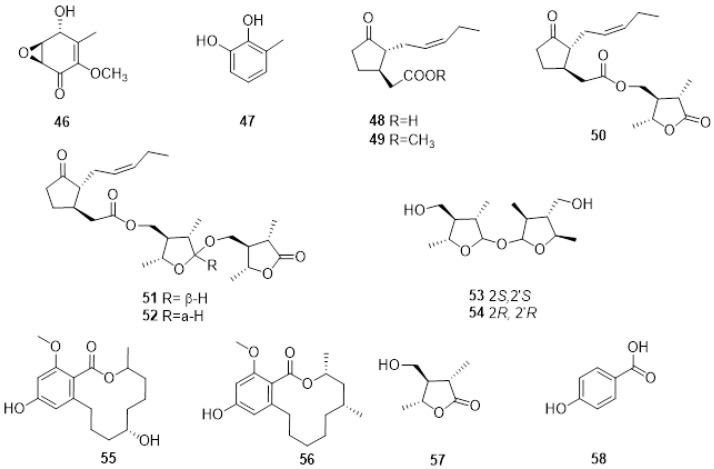
Structure of cyclobotryoxide (**46**), 3-methylcatechol (**47**), jasmonic acid (**48**), jasmonic acid methyl ester (**49**), lasiojasmonate A (**50**), lasiojasmonates B and C (**51**, **52**), lasiolactol A and B (**53**, **54**), botryosphaeriodiplodin (**55**), (5R)-5-hydroxylasiodiplodin (**56**), (3S,4R,5R)-4-hydroxymethyl-3,5-dimethyldihydro-2-furanone (**57**) and p-hydroxybenzoic acid (**58**).

**Figure 6 plants-11-03382-f006:**
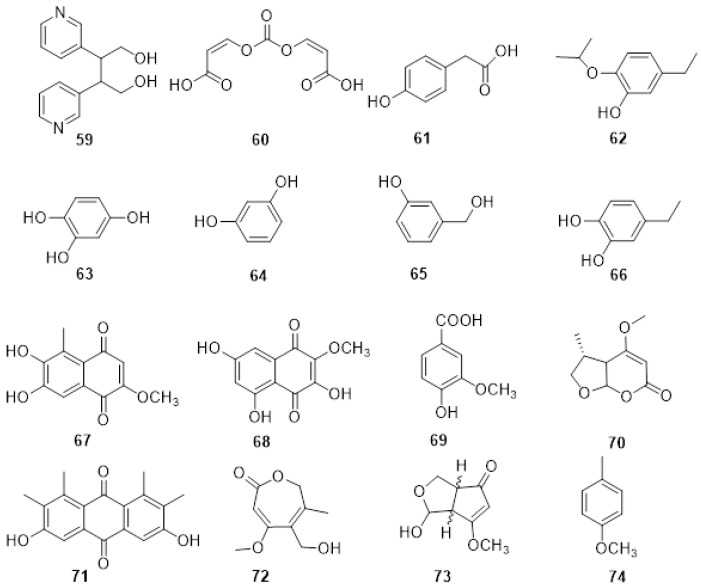
Structure of spencertoxin (**59**), spencer acid (**60**), 2-(4-hydroxyphenyl) acetic acid (**61**), 5-hydroxymethyl-2-isopropoxyphenol (**62**), benzene-1,2,4-triol (**63**), resorcinol (**64**), 3-(hydroxymethyl) phenol (**65**), protocatechuic alcohol (**66**), diploquinone A (**67**), diploquinone B (**68**), vanillic acid (**69**), luteopyroxin (**70**), neoanthraquinone (**71**), luteoxepinone (**72**), (±)-nigrosporione (**73**) and 4-cresol (**74**).

**Figure 7 plants-11-03382-f007:**
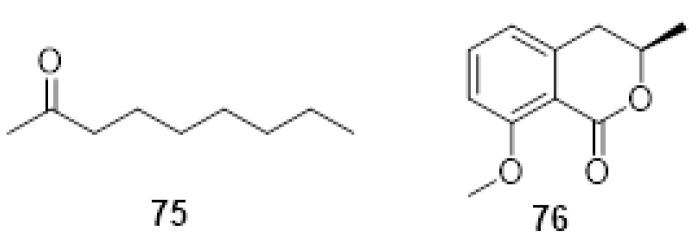
Structure of 2-nonanone (**75**), O-methylmellein (**76**).

**Table 1 plants-11-03382-t001:** Species of *Phaeoacremonium* and *Cadophora* involved in the Esca Complex.

Genus	Species	References
*Phaeoacremonium*	*P. angustius, P. alvesii,* *P. argentiniense, * *P. armeniacum, P. australiense,* *P. austroafricanum, * *P. canadense, P. cinereum,* *P. croatiense, P.globosum, * *P. hispanicum, P. hungaricum, P. inflatipes, P. italicum,* *P. iranianum, P. krajdenii, * *P. minimum, P. mortoniae,* *P. nordesticola, P. occidentale, * *P. roseum, P. scolyti, * *P. sicilianum, P. subulatum,* *P. tuscanum, P. venezuelense, P. viticola*	[16,42,43,45,46,47,48,49,50]
*Cadophora*	*Cad. luteo-olivacea, Cad. malorum, Cad. melinii, Cad. meredithiae, Cad. novi-eboraci, Cad. orchidicola, Cad. orientoamericana, Cad. spadicis, Cad. sabaouae, Cad. viticola*	[4,43,44,49,51]

**Table 2 plants-11-03382-t002:** Species of *Diatrypaceae* involved in Eutypa dieback.

Genus	Species	References
*Eutypa*	*E. laevata, E. consobrina,* *E. Cremea, E. lata, E. leptoplaca*	[65,66]
*Eutypella*	*Eu. Australiensis, Eu. Citricola, Eu. cryptovalsoidea, * *Eu. microtheca, * *Eutypella spp., Eu. vitis*	[4,63,66,67]
*Cryptosphaeria*	*C. lignyota, C. pullmanensis*	[68]
*Cryptovalsa*	*Cr. ampelina, Cr. rabenhorstii*	[68,69]
*Diatrype*	*D. brunneospora, * *D. oregonensis, D. stigma,* *Diatrype sp., D. whitmanensis*	[62,68]
*Diatrypella*	*Di. verrucaeformis, Di. vulgaris*	[62,63]

**Table 3 plants-11-03382-t003:** Species of Botryosphaeriaceae involved in Botryosphaeria dieback.

Genus	Species	References
*Botryosphaeria*	*B. dothidea*	[84,92]
*Diplodia*	*D. corticola, D. mutila, * *D. seriata, D.intermedia*	[4,84,89]
*Dothiorella*	*Do. americana, Do. neclivorem,* *Do. sarmentorum, Do.vidmadera, Do.vineagemmae*	[6,88,93,94]
*Lasiodiplodia*	*L. brasiliensis, L. citricola,* *L. crassipora, L. euphorbicola,* *L. exigua, L. hormozganensis,* *L. iraniensis, L.jatrophicola,* *L.laeliocattleyae, L.mahajangana* *L. mediterranea, L. missouriana, L. parva, L. pseudotheobromae,* *L. theobromae, L. viticola*	[84,89,90,95,96]
*Neoscytalidium*	*N. dimidiatum, N. hyalinum * *N. novaehollandiae*	[91,97,98]
*Neofusicoccum*	*N. australe, N. luteum, * *N. macroclavatum,* *N. mediterraneum, N. parvum, N. ribis, N. viticlavatum,* *N. vitifusiforme*	[51,84,92,93,99,100,101]
*Spencermartinsia*	*S. plurivora, S. viticola, * *S. westrale*	[6,88,89]

## Data Availability

Not applicable.

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
