# Peer review of "Phytotoxic Metabolites Produced by Fungi Involved in Grapevine Trunk Diseases: Progress, Challenges, and Opportunities"

_plants, 2022, doi:10.3390/plants11233382_

Round 1
Reviewer 1 Report
It would be preferable if You dedicated one paragraph on the managment GTDs.
Reviewer 2 Report
Dear Authors,
I found your review article entitled" Phytotoxic metabolites produced by fungi involved in Grapevine Trunk Diseases: progress, challenges, and opportunities" by Reveglia et al. interesting and revise important data about the phytotoxic metabolites produced by various fungi involved in GTD.
Overall, the paper is well documented and updated. It is also well written and presented. However, still some minor corrections need to be done and they are mentioned in the attached pdf file of the manuscript bellow.
Best Regards

Author Response
"Please see the attachment

Reviewer 3 Report
The review article “Phytotoxic metabolites produced by fungi involved in Grapevine Trunk Diseases: progress, challenges, and opportunities” by Reveglia et al., describes the chemical composition and biological characterization of the phytotoxic metabolites produced by fungi causing dieback of Eutypa and Botryosphaeria and Esca complex. Authors have also discussed the implications, challenges and opportunities to investigate, diagnose and control the grapevine trunk diseases (GTDs) in vineyards by employing multidisciplinary and cutting-edge technologies. In my opinion, this manuscript is informative and of scientific importance, however, the authors can make improvements in the manuscript in several places before the final publication.
The specific points and/or suggestions are given below:
1. Please carefully check the formatting of nomenclature to ensure consistency throughout the manuscript. For example, at page 16 N. parvum should be italicized in the sentence “This is the only report of PMs produced by N. parvum in grapevines showing Esca or BD in vineyards [97].”
2. While production of fungal metabolites is also affected by edaphic factors and climatic conditions, authors could briefly discuss how results from various cutting-edge techniques could be affected/misleading under such circumstances.
3. Please consider adding critical thoughts on how the development of vineyards disease symptoms could be differentiated/investigated in the presence of PMs, conducive climatic and physiological factors.
4. Multidisciplinary technologies exhibit great potential for the sustainable control of plant fungal diseases. Have you thought about integrating nanotechnology ? This is just one thought, if authors come across a relevant hypothesis/idea, I think it would further improve the critical aspect of the manuscript.
Reviewer 4 Report
The manuscript provides a detailed overview of the biological and chemical characterization of phytotoxic metabolites produced by fungi which causing Esca complex, Eutypa dieback and Botryosphaeria dieback. The manuscript is well organized and wrote, and it could be accepted after some minor revision, especially handwriting mistakes.
1. Page 8, “Molyneux and co-authors tudied three strains…by HPLC, GC MS” should be revised as “Molyneux and co-authors studied three strains…by HPLC, GC-MS”.
2. Page 11, I suggest the sentence of “…lipophilic low molecular weight phytotoxins were produced by the organic extracts…” should be changed to “…lipophilic low molecular weight phytotoxins were isolated from the organic extracts of the culture filtrates…”.
3. Page 13, “…are capable of producing jasmonic acid (48, Figure 5) a known plant hormone, and…” should be revised as “…are capable of producing jasmonic acid (48, Figure 5), a known plant hormone, and…”.
4. Page 16, “This is the only report of PMs produced by N. parvum in grapevines
showing Esca or BD in vineyards [97]”, N. parvum should be in italics. Similar problems in the whole manuscript should be revised.
5. Page 18, “To quantify scytalone (1) and isosclerone (2) a sensitive…” could be revised as “To quantify scytalone (1) and isosclerone (2), a sensitive…”.
